# Symmetry-broken Josephson junctions and superconducting diodes in magic-angle twisted bilayer graphene

J. Díez-Mérida [1], A. Díez-Carlón [1], S. Y. Yang[1], Y.-M. Xie [2], X.-J. Gao[2],
J. Senior [3], K. Watanabe [4], T. Taniguchi [4], X. Lu [1], A. P. Higginbotham [3],
K. T. Law [2] & Dmitri K. Efetov [1] ✉

The coexistence of gate-tunable superconducting, magnetic and topological orders in magic-angle twisted bilayer graphene provides opportunities for the creation of hybrid Josephson junctions. Here we report the fabrication of gate-defined symmetry-broken Josephson junctions in magic-angle twisted bilayer graphene, where the weak link is gate-tuned close to the correlated insulator state with a moiré filling factor of $\upsilon = -2$. We observe a phase-shifted and asymmetric Fraunhofer pattern with a pronounced magnetic hysteresis. Our theoretical calculations of the junction weak link—with valley polarization and orbital magnetization—explain most of these unconventional features. The effects persist up to the critical temperature of 3.5 K, with magnetic hysteresis observed below 800 mK. We show how the combination of magnetization and its current-induced magnetization switching allows us to realise a programmable zero-field superconducting diode. Our results represent a major advance towards the creation of future superconducting quantum electronic devices.

Electronic coupling between materials with competing ground states can lead to the creation of exotic electronic phases. Of particular interest are hetero-junctions between superconductors, magnets, and topological insulators, where especially Josephson junctions (JJ) thereof have attracted formidable attention. Magnetic JJs allow spintronic applications through the creation of spin-filters[1–5], spin-triplet supercurrent[6–8], and π junctions[9–12], whereas topological JJs allow applications in quantum information processing and in lossless electronics, through the creation of 4π junctions[13–15], superconducting diodes[16–18], and Majorana bounds states[19]. One major difficulty in the creation of such JJs lies in the engineering of ultra-clean interfaces between the different material species, which is needed for efficient coupling between different phases.

A single, two-dimensional material that would host all of these emergent phases at once, would overcome these issues. It would permit to induce gate-defined, ultra-clean homojunction between all the different phases, and so open up a new avenue for the creation of a new generation of superconducting electronics. The recently discovered quantum phases in the flat bands of $\theta_m$-1.1° magic-angle twisted bilayer graphene (MATBG) include correlated insulators (CI)[20–24], superconductors (SC)[22,23,25–27], orbital magnets (OM)[23,28,29], and interaction induced correlated Chern insulators (CCI)[30–34].

Here, we demonstrate the creation of a symmetry-broken JJ in a locally gated MATBG device, when the weak link is set close to half-filling of the hole band. Remarkably, due to MATBG's two-dimensionality and ultra-low carrier density, it is possible to use an electrostatic gate to tune between the different phases, and induce reversible transitions between the SC, CCI, and OM phases[35]. Although there have already been reports on the creation of MATBG JJs through the use of local gates[36,37], JJs with the supercurrent mediated by the

[1]ICFO - Institut de Ciencies Fotoniques, The Barcelona Institute of Science and Technology, Castelldefels, Barcelona 08860, Spain. [2]Department of Physics, Hong Kong University of Science and Technology, Clear Water Bay, Hong Kong, China. [3]IST Austria, Am Campus 1, 3400 Klosterneuburg, Austria. [4]National Institute for Materials Science, 1-1 Namiki, Tsukuba 305-0044, Japan. ✉e-mail: dmitri.efetov@physik.lmu.de

strongly correlated magnetic or topological weak links have so far not been achieved.

## Results

### Gate-defined JJ

The device consists of a van der Waals (vdW) hetero-structure of gra-phite/hBN/MATBG/hBN/graphite, as shown in Fig. 1a. Metallic graphite layers are capacitively coupled to the MATBG, through the insulating hexagonal boron nitride layers (hBN) of ~10 nm thickness. The carrier density in the MATBG sheet $n$ is electrostatically tuned by both top and bottom gates $n = C_{BG}V_{BG} + C_{TG}V_{TG}$ where $(C_{BG}, C_{TG})$ are the respective capacitances and $(V_{BG}, V_{TG})$ gate voltages. In the center of the device, the top graphite layer is separated by a narrow channel of length $d = 150$ nm, which creates a region in the MATBG that is almost not coupled to the top gate, and whose carrier density is mainly set by the back gate voltage $n_J \sim C_{BG}V_{BG}$. Hence, by applying different values of $V_{BG}$ and $V_{TG}$, it is possible to locally vary the carrier concentration in the channel region, which allows creating gate-defined junctions of length $d_J \sim 100$ nm in the MATBG, as is confirmed by electrostatic simulations and is highlighted in Fig. 1b.

Figure 1c shows the 4-terminal longitudinal resistance $R_{xx}$ as a function of $V_{BG}$ ($V_{TG} = 0$ V) at base temperature $T = 35$ mK, and for different perpendicular magnetic fields $B$. From Hall and quantum magneto-oscillation measurements (see Methods) we extract a twist-angle of $\theta = 1.11° \pm 0.02°$. We observe well-pronounced CI states, which give rise to peaks of high resistance at integer electron and hole fillings of the moiré unit cell, $\upsilon = +1, \pm2,$ and $+3$, as well as a SC state on the hole-doped side of $\upsilon = -2$, with a critical temperature $T_c \sim 3.5$ K (see Supplementary Fig. 3). Overall the device shows a phase diagram which is very similar to previous reports of hBN non-aligned MATBG devices[22,23,25,26],

which is confirmed by examining the crystallographic edges (see Supplementary Fig. 1).

In order to create a JJ in the device, we control both $(V_{BG}, V_{TG})$ to tune the $n$ to $n_{sc} = -1.72 \times 10^{12}$ cm$^{-2}$, where the SC state is at optimal doping. By further changing $(V_{BG}, V_{TG})$ following the relation $\Delta V_{BG} = -(C_{TG}/C_{BG})/\Delta V_{TG}$, $n$ can be kept constant, while the carrier density in the junction $n_J$ is continuously tuned (see Supplementary Information for detailed dual-gate maps). This allows to tune the junction region from a metallic (N) to a SC and into a CI state. As the length of the junction is in the order of magnitude of the coherence length of the SC $d_J \sim \xi \sim 100$ nm (see Supplementary Fig. 3), it is possible to proximitize the junction and to create a JJ[38].

Figure 1d shows the differential longitudinal resistance $dV_{xx}/dI$ vs. source-drain current $I$ and as a function of $n_J$, for a range of fillings which is centered around the SC state $-3 < \upsilon < -2$. The upper panel shows the corresponding $R_{xx}$ vs. $n_J$ measurement which demonstrates the density ranges of the N, SC and CI states. For $n_J = n_{sc}$, the device is uniformly in the SC state, and forms a SC/SC/SC junction, with an $I_c > 200$ nA. However, when $n_J$ is tuned away from this point a SC/SC'/SC junction is created, where we observe a second set of coherence peaks with reduced $I_c$. For density ranges close to $\upsilon = -3$, $-2.2 \times 10^{12}$ cm$^{-2} \lesssim n_J \lesssim -1.86 \times 10^{12}$ cm$^{-2}$, and close to $\upsilon = -2$, $-1.58 \times 10^{12}$ cm$^{-2} \gtrsim n_J \gtrsim -1.5 \times 10^{12}$ cm$^{-2}$, the junction region is not intrinsically superconducting, but we still observe a supercurrent, which hints at the creation of a JJ. While close to $\upsilon = -3$ the junction is metallic and a SC/N/SC is formed, close to $\upsilon = -2$ the junction is in the vicinity of the correlated insulator state making a SC/CI'/SC. Beyond these density ranges we do not observe a supercurrent, however, distinct superconducting non-linearities remain, which are in-line with Andreev reflections at the SC interfaces[39].

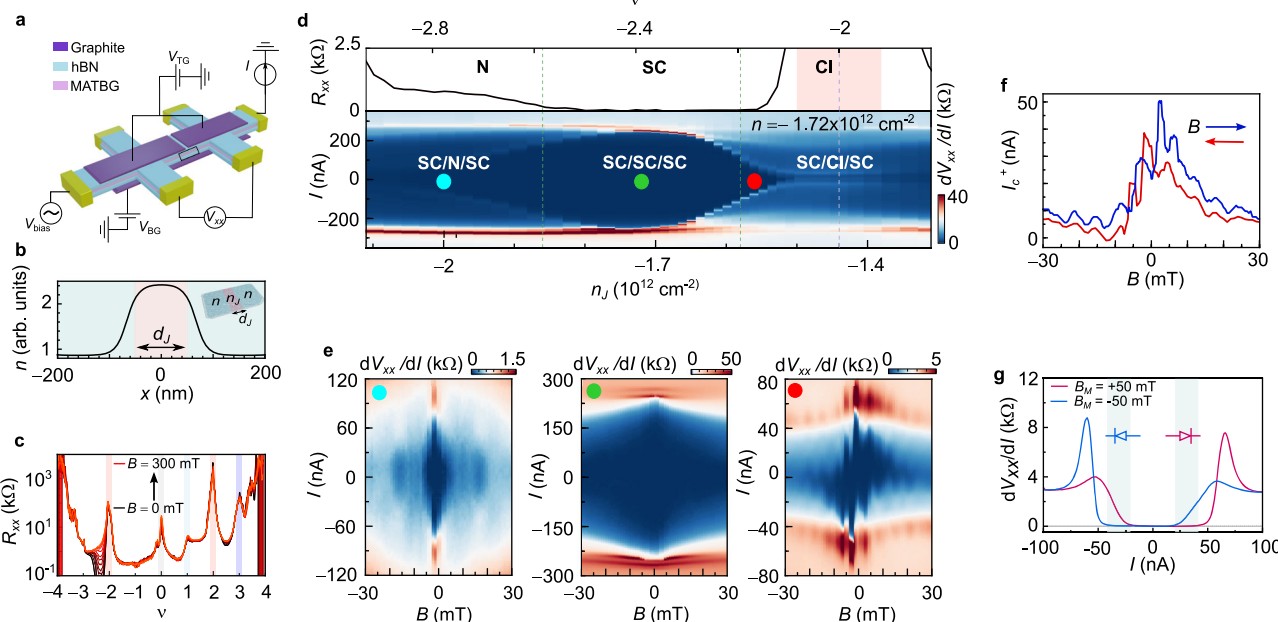

**Fig. 1 | Gate-tunable JJ in MATBG. a** Schematic of the measured device and measuring circuit, where $V_{bias}$ is the source voltage, $I$ is the current through the device, $V_{xx}$ the voltage drop between the measurement probes, and $V_{BG}$ ($V_{TG}$) correspond to the back (top) gate voltage. The top graphite gates are separated by 150 nm. **b** Electrostatic simulation profile of carrier density $n$ vs. position $x$, setting $n$ and carrier density in the junction $n_J$ at different values with a junction length $d_J$ of 100 nm. The inset shows a schematic of the MATBG JJ with two distinct regions created by the gating structure. **c** Four terminal longitudinal resistance $R_{xx}$ vs. filling factor $\upsilon$ at different out-of-plane magnetic fields $B$ from 0 mT (black curve) to 300 mT (red curve). **d** (Top) Magnification of **c** around the superconducting state $-3 < \upsilon < -1.8$, where we define three distinct regions with metallic (N),

superconducting (SC), and correlated insulator (CI) behavior. (Bottom) $dV_{xx}/dI$ vs. $I$ at different $n_J$, keeping $n = -1.72 \times 10^{12}$ cm$^{-2}$ in the SC state. Dashed green vertical lines mark the position where $n_J$ is no longer in the SC state. **e** Fraunhofer patterns measured at (left) $n_J = -2 \times 10^{12}$ cm$^{-2}$ (SC/N/SC), (center) $-1.72 \times 10^{12}$ cm$^{-2}$ (SC/SC/SC), and (right) $-1.56 \times 10^{12}$ cm$^{-2}$ (close to SC/CI/SC), respectively. The color dots show the corresponding $n_J$ positions in the $dV_{xx}/dI$ vs. $I$ map in **d** bottom. **f** Positive critical current $I_c^+$ vs. $B$ with $B$ sweeping up (blue) and down (red). **g** $dV_{xx}/dI$ vs. $I$ at $B = 0$ mT after applying a pre-magnetizing field $B_M = +50$ and $-50$ mT for the red and blue curve. The shaded gray regions mark the values of current at which the diode behavior is observed.

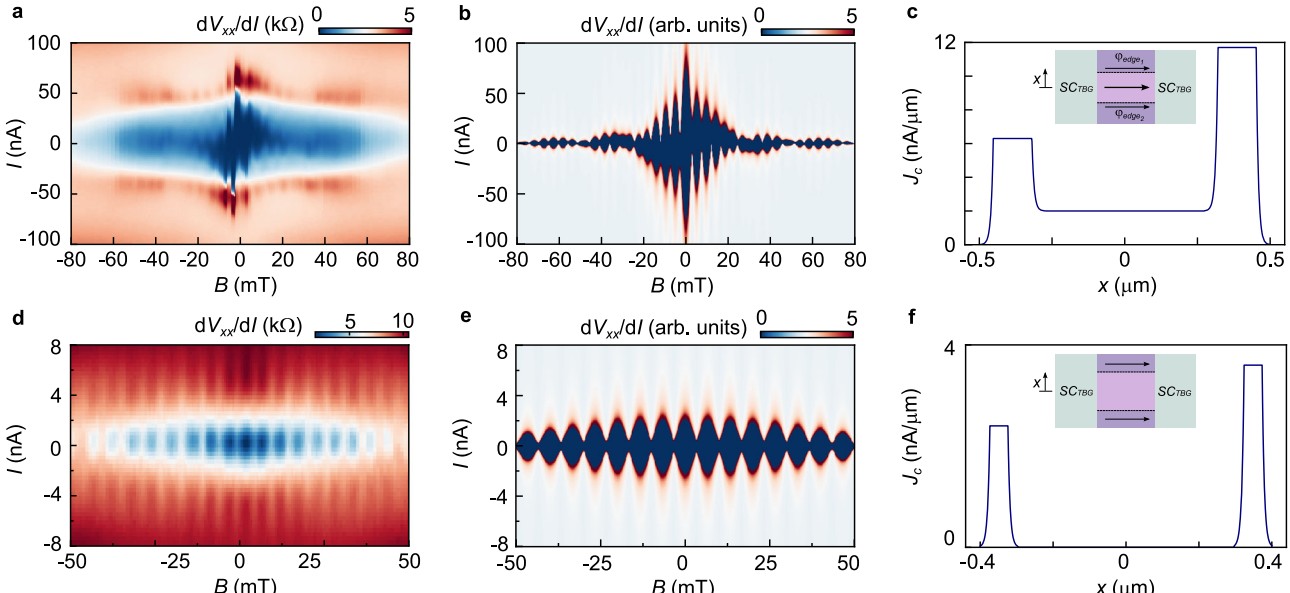

**Fig. 2 | Fraunhofer patterns emerging from edge state supercurrent of device A and B. a** Fraunhofer pattern at the SC/CI′/SC position of sample A up to higher out-of-plane magnetic field $B$. A revival of the oscillations is observed after 30 mT. **b** Calculated $I_c(B)$ behavior based on the current density distribution shown in **c**. A combination of edge and bulk supercurrent with non-symmetric edges having different phases $\varphi_{edge,n}$, where $n = 1, 2$ for the top and bottom edge respectively, such that $\varphi_{edge1}(I > 0) = -\pi/4$, $\varphi_{edge1}(I < 0) = \pi/4$ and $\varphi_{edge2} = -\varphi_{edge1}$ give rise to a qualitatively similar pattern as measured in the experiment. **c** Current density distribution combining edge and bulk supercurrent. The inset shows a cartoon displaying the superconducting TBG forming the JJ ($SC_{TBG}$) in light green and the weak link with bulk and edge contribution in purple and violet, as well as the different phases carried by the edges to calculate the pattern in **b**. **d** Fraunhofer pattern at the SC/CI′/SC position of sample B. The pattern resembles a pattern coming from purely edge supercurrent. **e** Calculated $I_c(B)$ behavior based on the current density distribution shown in **f**, in which all the supercurrent is carried by the edges. **f** Current density distribution with only edge conduction. The inset shows a cartoon in which the current is only carried by edge states, without acquiring any extra phase.

## Unconventional Fraunhofer pattern and superconducting diode

We further analyze the various gate-induced junctions by applying an out-of-plane magnetic field $B$, where Fig. 1e shows the color maps of $dV_{xx}/dI$ vs. $I$ vs. $B$. For the uniform SC/SC/SC junction (center) we observe a diamond-shaped dependence, which is symmetric with the inversion of the current $I_c^+(B^+) = I_c^-(B^+)$ and the $B$-field directions $I_c^+(B^+) = I_c^+(B^-)$, where $I_c^+(I_c^-)$ and $B^+(B^-)$ correspond to the positive (negative) critical current and field. This behavior is in-line with previous reports on SC states in MATBG[22,23,25–27], with a critical magnetic field $B_{c2} \sim 120$ mT. The absence of Fraunhofer oscillations confirms the uniformity of the junction and the absence of a JJ. For both, the SC/N/SC (left) and SC/CI′/SC (right) junctions (taken at $n_J$ as marked in Fig. 1d) we observe clear Fraunhofer oscillations with oscillation periods that are consistent with expectations for one flux quantum through the junction (see Supplementary Information) which unambiguously prove the formation of a gate-defined JJ[37,40]. Here the SC/N/SC JJ displays a typical Fraunhofer pattern, which obeys the same symmetries along the $I$ and $B$ directions as the pristine SC state. In stark contrast to this, the SC/CI′/SC JJ shows a very unusual Fraunhofer pattern, which is not symmetric with inversion of the current $I_c^+(B^+) \neq I_c^-(B^+)$ and the $B$-field directions $I_c^+(B^+) \neq I_c^+(B^-)$, which indicates time-reversal symmetry breaking. These asymmetries are well seen in the line cuts in Fig. 1f, g which show $I_c^+$ vs. $B$ and $dV_{xx}/dI$ vs. $I$ measurements respectively. Most strikingly, for both measurements we observe a hysteresis as a function of $B$-field direction. A direct consequence of the hysteresis in the $dV_{xx}/dI$ vs. $I$ is its non-reciprocal transport. This is demonstrated in Fig. 1g, where for a fixed current value $|I| \sim 10–50$ nA the device can be superconducting in one current direction, while highly resistive in the other. This behavior enables the creation of a superconducting diode, which is the superconducting analog of a p-n junction, and is highly sought after as a building block for superconducting electronics. Since the magnetization direction can be switched by a small field $B_M$ (red and blue lines in Fig. 1g), the polarity of the current asymmetry can be switched, and the direction of the diode reversed, making it so programmable (see Supplementary Fig. 11). We now focus on the Fraunhofer pattern at the SC/CI′/SC position. First, we study the behavior to higher magnetic fields, where we observe a revival of the oscillations and the SC after a field of ±30 mT (Fig. 2a), where the oscillations have completely decay and then reappear again. The double periodicity suggests the presence of edge states giving rise to a SQUID-like type of behavior[41]. In order to better understand the origin of the signal, we calculate a Fraunhofer pattern corresponding to a given current density distribution in real space $I_c(\beta) = |\int_{-\infty}^{\infty} dx \, J_s(x) \, e^{i\beta x}|$ where $\beta(B)$ is a normalized field along the length of the JJ and $J_s(x)$ is the real space current density distribution[42] (see Supplementary Information for details). By having a $J_s(x)$ combining edge and bulk states we can obtain a pattern similar to the one observed in experiment (Fig. 2b, c). Starting from this combined bulk and edge supercurrent, we can also simulate the other features of the pattern, mainly that the central lobes do not reach zero at the oscillation period and the asymmetries with respect to the field and current directions. The fact that the lobes do not reach zero can be understood by having asymmetric edges, which we can simply simulate by attributing a different critical current to each edge. The asymmetries in the field direction ($I_c^+(B^+) \neq I_c^+(B^-)$), however, require that the edges carry an extra phase different from the bulk. This phase acts as an effective magnetic field in the $\beta$ parameter, substituting the field $B$ as $B = B_{ext} + \varphi$, where $B_{ext}$ is the externally applied field and $\varphi$ the extra phase acquired by the sample (see Supplementary Information for details). Finally, in order to obtain the measured current-field asymmetry ($I_c^+(B^+) \neq I_c^-(B^+)$) these phases have to change sign for opposite current directions. The final pattern plotted in Fig. 2b is obtained by making $\varphi_{edge1} \neq \varphi_{edge2}$ and $sgn(\varphi(I^+)) = -sgn(\varphi(I^-))$. A second sample with twist-angle 1.04 ± 0.02° has also been measured (device B). The $I_c(B)$ behavior of the new sample at the SC/CI′/SC position is shown in Fig. 2d. In this case we do not observe any asymmetries or hysteretic behavior, but the sample displays clear SQUID-like oscillations. We can again model the pattern, in this case having a supercurrent which is purely carried by

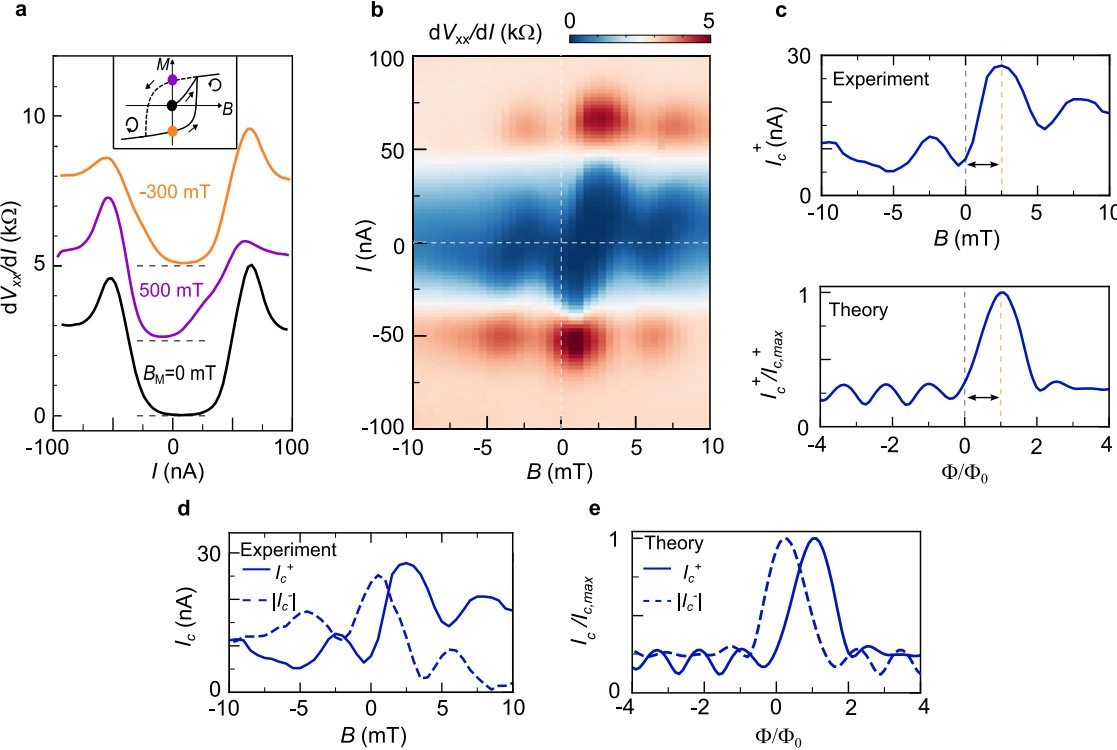

**Fig. 3 | JJ with orbital magnetism. a** $dV_{xx}/dI$ vs. $I$ measured at $B = 0$ mT and $T = 800$ mK right after cooldown (black) and after the sample has been subjected to two opposing pre-magnetizing fields $B_M$. The curves are vertically shifted by 2.5 kΩ each for clarity. The inset shows a schematic of magnetization $M$ vs. $B$. The colored dots correspond to the magnetic states in which the different $dV_{xx}/dI$ vs. $I$ curves were taken and the arrows describe the direction in which the field is swept. **b** Fraunhofer pattern with $n_J = -1.56 \times 10^{12}$ cm$^{-2}$ measured at 800 mK. The white dashed lines mark the 0 current and 0 field positions. **c** (Top) Positive critical current $I_c^+$ vs. $B$ at 800 mK. The vertical dashed lines remark the shift of the $I_c^+$

maximum from zero field. (Bottom) Theoretical $I_c^+$ vs. magnetic flux ($\Phi$) normalized by the flux quantum ($\Phi_0$) calculated for a MATBG JJ with a valley-polarized $\upsilon = -2$ state as the weak link. The pattern has been shifted by $+\Phi_0$ to compare with the experiment. **d** Experimental $I_c^+$ and $|I_c^-|$ vs. $B$, extracted from **b**. Reversing the current direction inverts the line-shape of the curve and changes the shift in magnetic field. **e** Theoretical $I_c^+$ and $|I_c^-|$ vs. $\Phi$ for a MATBG JJ with a valley-polarized $\upsilon = -2$ state as the weak link. To compare with the experiment, a shift of $+\Phi_0$ and $+0.2\Phi_0$ was added to $I_c^+$ and $|I_c^-|$, respectively.

the edge states, hinting that in device B the CI state has a more insulating bulk than for device A. For both devices we observe how in the SC/CI′/SC configuration edge states play an important role carrying the supercurrent.

**Magnetic JJ**

Next, we analyze the magnetic signatures of device A. We first examine the Fraunhofer pattern with the parameters of Fig. 1e (right) at an elevated temperature of $T = 800$ mK, where the hysteretic behavior at weak magnetic field has not yet developed (Fig. 3a–c). In this regime, the Fraunhofer pattern has the following highly unconventional features: 1. The central peak of the Fraunhofer pattern is shifted from $B = 0$ to a value of $B \sim 2.5$ mT; 2. The Fraunhofer pattern is highly asymmetric with respect to the central peak; 3. The critical current $I_c^+$ does not vanish as a function of $B$; 4. Even more strikingly, when the current direction is reversed, a different Fraunhofer pattern is observed, and the central peak is shifted as shown in Fig. 3d. At $B = 0$, for example, the critical current is dramatically different for currents flowing in opposite directions. 5. The critical current shows a hysteresis and the directional dependence of the critical current appears only when the system is pre-magnetized by an external magnetic field larger than a coercive field of ~300 mT (purple and orange lines in Fig. 3a). As none of these features are observed in the SC/SC/SC and the SC/N/SC junctions, we suggest that the CI state in the middle of the JJ is an unconventional insulating state responsible for the observed Fraunhofer pattern. Qualitatively, the shift of the central peak, the breaking of the time-reversal symmetry condition $I_c^+(B^+) = I_c^+(B^-)$ and the hysteresis behavior all suggest that time-reversal symmetry is broken and

there is a spontaneous net magnetic flux which is responsible to move the position of the central peak away from the $B = 0$ position. Furthermore, the observed behavior in Fig. 2 indicates that edge states play an important role in carrying the supercurrent. It is important to note that the observed unconventional Fraunhofer patterns are highly reproducible, i.e. we do not observe significant changes in the patterns after several thermodynamic cycles of warming up and cooling down the sample.

The question is: Which microscopic state of MATBG near $\upsilon = -2$ can explain these? We propose below that the observed experimental features are consistent with the assumption that the CI is an interaction induced valley-polarized state with net orbital magnetization.

This state has been previously identified at slightly elevated magnetic fields $B > 300$ mT[26], which is in good agreement with the observed coercive field of the JJ. Moreover, the orbital magnetic moment of this state is huge ~6 $\mu_B$ (Bohr magneton)[43] and produces an out-of-plane magnetic field of $B \sim 3$ mT (see Supplementary Information for derivation). This is consistent with the experimentally obtained phase shift of $\Delta B \sim 2.5$ mT. The phase shift of the Fraunhofer pattern survives up to the critical temperature of the JJ of $T_c \sim 1$ K, and is comparable to the Curie temperature of previously observed orbital magnetic states in hBN aligned[28,29] and non-aligned MATBG[23,26] as well as in twisted mono-bi graphene[44,45]. Finally, the valley-polarized state with orbital magnetization is characterized by the presence of edge states, which would arise as observed in the Fraunhofer patterns of Fig. 2.

To further support this hypothesis, we construct a MATBG-based JJ model by assuming the CI in the central region to be a

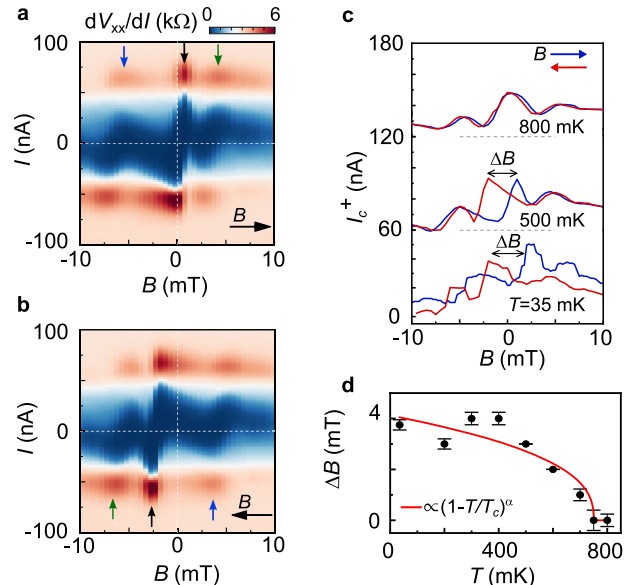

**Fig. 4 | Evolution of magnetic hysteresis with temperature. a, b** Fraunhofer patterns measured at 500 mK with field sweeping up (**a**) and down (**b**) as marked with the black arrows. The white dashed lines mark the 0 current and 0 field positions. The colored arrows highlight a period change in the pattern and the fact that by rotating **a** by 180° one would get the periodicity of **b**. **c** $I_c^+$ extracted from the Fraunhofer patterns with the magnetic field sweeping up (blue) and down (red) at 800, 500, and 35 mK. The curves are vertically shifted by 60 nA each for clarity. **d** Extracted $\Delta B$ vs. temperature $T$ for $I_c^+$. The red curve is a fit to the Curie–Bloch equation $(1-T/T_c)^\alpha$ with fitting parameters $T_c$ ~750 ± 25 mK and $\alpha$ ~0.4 ± 0.05. The error bars are defined as the standard deviation of the extracted $I_c$ values as explained in the SI.

valley-polarized state with net Chern number $C = -2$ at filling factor $v = -2$, while the Chern bands are partially filled (see Supplementary Information for details). The superconducting part of the JJ is assumed to be a fully gapped superconductor with s-wave pairing. The theory clearly reproduces the asymmetry with respect to the central peak of the unconventional Fraunhofer pattern (Fig. 3c). Unlike in the case of a conventional Fraunhofer pattern, it is asymmetric with respect to the $B$-field direction, where $I_c(B^+) > I_c(B^-)$. We found that removing the $C_2T$ breaking terms will make the bands topologically trivial with no Berry curvatures nor net orbital magnetic moments. In this case, a standard Fraunhofer pattern is obtained (see Supplementary Information). This suggests that this behavior is a direct consequence of the electronic ground state near $v = -2$ carrying orbital magnetization. We can then simulate the behavior of device B by setting the chemical potential into the gap of the CI, equivalent to having a more insulating bulk (see Supplementary Information), in which case the current is solely carried by edge states, consistent with the results shown in Fig. 2. Therefore, the main features of both devices can be captured within the same model.

To explain the directional dependence of the critical current in Fig. 3d, one extra assumption is needed. Namely, the current $I$ ~ 10 nA can induce orbital magnetization switching similar to the current-induced orbital magnetization switching, which is observed at a filling of $v = 3$ in MATBG[28,29,44,46–48]. In other words, a small current can overcome the free energy barrier between two degenerate orbital magnetization states of the CI. With this assumption, which is further motivated by the model of Fig. 2, the directional dependence of the critical current is well explained (Fig. 3e). In the case of device B, the fact that there is no bulk current and the $I_c$ is an order of magnitude smaller (~5 nA vs. 80 nA), could be the reason why no asymmetry is observed. However, further theoretical study is needed to understand the current-induced orbital magnetization switching in this $C = -2$ state.

The Fraunhofer pattern of device A at low temperatures is even more intriguing. Figure 4a, b shows it for $T = 500$ mK, where it is measured by sweeping the $B$-field up (a) and down (b). Strikingly, both Fraunhofer patterns show a phase jump (marked by an arrow), which was not observed at higher temperatures. Comparing the two Fraunhofer patterns, one notices that they are phase-shifted, and overall symmetric with respect to the reversal of the current and $B$-field directions, $I_c^+(B^{+,\rightarrow}) \sim I_c^-(B^{-,\leftarrow})$. Its phase jump is hysteretic and occurs at different $B$-fields for the up ($B^\rightarrow$) and down($B^\leftarrow$) sweeps. Such $B$-field hysteresis is better seen in the line cuts in Fig. 4c, which shows the $I_c^+(B)$ for both field sweeping directions at $T = 800$, 500, and 35 mK. Here we define $\Delta B$ as the difference between the maxima of the $I_c^+(B^+)$ and $I_c^+(B^-)$ sweeps. If we understand this hysteresis as the magnetization of the sample and plot its temperature dependence, we can fit it with a Curie–Bloch equation[49] $\Delta B \sim (1-T/T_c)^\alpha$ (Fig. 4d) obtaining a Curie temperature $T_c$ ~750 ± 25 mK and $\alpha$ ~ 0.4 ± 0.05.

## Discussion

Both the hysteresis of $I_c(B)$ and the phase jumps are prominent characteristics of ferromagnetic JJs[3–5]. The hysteresis is induced by a switching of the magnetic orientation and the phase jumps are due to the presence of domains switching at different field values. While the $I$-$B$ asymmetry, indicative of orbital magnetism, continues to be present in the Fraunhofer pattern, the low $T$ hysteretic features cannot be fully explained by it. These appear at a lower temperature and require two orders of magnitude lower switching field $|B_M| \geq 3$ mT than observed for the valley-polarized state. Therefore, a further theoretical explanation is needed to explain these lower $T$ features which have a clear distinct behavior.

For now, we can just propose possible scenarios based on the data. A first scenario would be to have both spin and valley polarization, for example, by having a partially valley-polarized state, in which both the spin and valley flavors have a population imbalance. Such states have been studied recently as possibilities to explain magnetic signals observed at $v = -2$ and had been previously discussed in literature[50,51]. In this case, the valley and spin polarization could have different energy scales, being responsible for the observed signals. Another alternative would be to have domains of different magnetic behavior as has been recently observed in a SQUID on tip experiment[52]. In this case, there could be domains all of orbital origin or a combination of domains of orbital and spin origin. In the latter case the spin and orbital domains could behave differently while, in the former case, the different behavior could be coming from domains of different sizes or domains having a different type of magnetic behavior as was observed in ref. 52. Considering the phase jumps in the data at low $T$ and the modeling of the current density with opposing phases on both edges, the domain picture might be a more likely scenario. However, a definite proof of the origin of these signals cannot be drawn from the present study.

To summarize, we have proved that time-reversal symmetry-broken states can coexist with superconductivity in a single MATBG device. The zero-field coexistence and gate tunability of the magnetic and topological phases with superconductors in MATBG presents a remarkable opportunity to electronically hybridize these phases through engineering of complex gate-induced junctions. This will lead to the creation of ever more complex quantum phases based on the MATBG platform. Also, the so-created JJs can shed new light on the underlying ground states of MATBG, as the JJ probes much smaller areas than traditional transport experiments and are highly sensitive to magnetic fields.

## Methods

### Device fabrication

The MATBG devices are fabricated using a cut-and-stack technique. All flakes were first exfoliated in a Si/SiO$_2$ (285 nm) substrate and later

picked up using a polycarbonate (PC)/polydimethylsiloxane (PDMS) stamp. All the layers were picked up at a temperature of ~100 °C. The graphene is initially cut with an AFM tip, to avoid strain during the pick-up process. The PC/PDMS stamp was used to pick-up first the top graphite layer, the first hBN, and the first graphene layer. Before picking up the second graphene layer, the stage is rotated by an angle of 1.1–1.2°. Finally, the bottom hBN and bottom graphite gates were picked up. The finalized stack is dropped on a Si/SiO₂ substrate by melting the PC at 180 °C. The resulting stack is etched into a Hall bar with $CHF_3/O_2$ and a 1D contact is formed by evaporating Cr (5 nm)/Au (50 nm). The narrow channel of ~150 nm in the top gate is etched with $O_2$. Before etching the top gate, the device was characterized at $T = 35$ mK to identify the pair of contacts closest to the magic-angle ($\theta \sim 1.1°$). The junction was made in between this pair of contacts.

## Measurements

Transport measurements were carried out in a dilution refrigerator (Bluefors SD250) with a base temperature of 20 mK. Standard low-frequency lock-in techniques (Stanford Research SR860 amplifiers) were used to measure $R_{xx}$ with an excitation current of 10 nA at a frequency of 13.11 Hz. For the $dV_{xx}/dI$ measurements the excitation current was reduced to 1 nA. The d. c. bias current was applied through a 1/100 divider and a 1 MΩ resistor before combining it with the a.c. excitation. Keithley 2400 Source-meters were used to control the gates as well as serve as the source for the DC current. The measured $dV_{xx}/dI$ signals were filtered and amplified by voltage-preamplifiers SR560 before entering the lock-in amplifiers.

## Twist-angle extraction

The twist angle is extracted from the phase diagrams shown in Supplementary Fig. 2. The carrier density corresponding to a fully filled superlattice unit cell is extracted to be $n_s = (2.88 \pm 0.1) \times 10^{12}$ cm⁻². By applying the relation $n_s = 8\theta^2/\sqrt{3}a^2$, where $a = 0.246$ nm is the graphene lattice constant, we extract a twist angle $\theta = 1.11° \pm 0.02°$.

## Data availability

The data that support the findings of this study are available at the public repository Zenodo under accession code: https://zenodo.org/record/7774670. Other data which might be relevant is available from the corresponding author upon request.

## Code availability

The code that supports the findings of this study is available from the corresponding author upon request.

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

## Acknowledgements

We are grateful for the fruitful discussions with Allan MacDonald and Andrei Bernevig. D.K.E. acknowledges support from the Ministry of Economy and Competitiveness of Spain through the "Severo Ochoa" program for Centers of Excellence in R&D (SE5-0522), Fundació Privada Cellex, Fundació Privada Mir-Puig, the Generalitat de Catalunya through the CERCA program, funding from the European Research Council (ERC) under the European Union's Horizon 2020 research and innovation program (grant agreement no. 852927)" and the La Caixa Foundation. K.T.L. acknowledges the support of the Ministry of Science and Technology of China and the HKRGC through grants MOST20SC04, C6025-19G, 16310219, 16309718, and 16310520. J.D.M. acknowledges support from the INPhINIT 'la Caixa' Foundation (ID 100010434) fellowship program (LCF/BQ/DI19/11730021). Y.M.X. acknowledges the support of HKRGC through Grant No. PDFS2223-6S01.

## Author contributions

D.K.E. and X.L. conceived and designed the experiments; J.D.M., A.D.C., and X.L. fabricated the devices and performed the measurements; J.D.M., A.D.C., S.Y.Y., D.K.E., Y.M.X, X.G., and K.T.L. analyzed the data; Y.M.X., X.J.G., and K.T.L. performed the theoretical analysis; T.T. and K.W. contributed materials; J.S., A.P.H., and D.K.E. supported the experiments: J.D.M., A.D.C., S.Y.Y., D.K.E., Y.M.X., and K.T.L. wrote the paper.

## Competing interests

The authors declare no competing interests.
