## [Peer Review File · Nature Communications]

REVIEWER COMMENTS

Reviewer #1 (Remarks to the Author):

The authors have addressed all of my original concerns. They have substantially strengthened their manuscript both with improved theoretical analysis and the inclusion of a second device. I am happy to recommend publication in Nature Communications.

Reviewer #2 (Remarks to the Author):

After thoroughly reviewing the authors' responses to the comments from the referees, I am pleased to say that all of the concerns raised have been satisfactorily addressed in the revised manuscript. I have also checked the replies to the other referees' comments and found that the answers provided are thorough and convincing. Particularly, the authors have clarified the confusion about the Chern band and spin polarization around $\nu=-2$ filling. The authors also acknowledged that an unambiguous determination of the nature of the insulating state based on the data available in the current experiment is not possible. Nevertheless, the observations of work will certainly shed light on the nature of the correlated insulating state, and impose strong constraints on future theory work. In addition, the authors have also fabricated more devices to reproduce the claimed observation, which makes the current study more reliable. The authors have done an excellent job in addressing the comments and making the necessary revisions to the main manuscript. Based on my review, I am confident in recommending the manuscript for publication.

Reviewer #3 (Remarks to the Author):

Questions:

Thanks to the authors' reply. Overall, I am fine with some questions from the earlier manuscript. However, before publishing, there are still some issues.

Comment:

1. One of the questions from the referee is whether the author should observe the quantum anomalous Hall effect due to the valley polarization with a Chern number of -2 for the correlated insulating state. This effect could be greatly lifted due to the geometry effect. In a narrow junction geometry, the R_{xx} and R_{xy} often have severe mixing. Indeed, if one considers the Hall geometry, the anomalous Hall effect may be expected. However, it will be hard to see in a gated junction geometry. See the following reference: Williams, J. R., Abanin, D. A., DiCarlo, L., Levitov, L. S., & Marcus, C. M. (2009). Quantum Hall conductance of two-terminal graphene devices. *Physical Review B*, 80(4), 045408.
2. The new device B has relatively weak superconductivity as shown in the SI. The resistance doesn't reach zero at zero fields. This may be hard to reproduce the CI state in device A. It is also the reason that interference shows a SQUID pattern due to the poor superconducting properties of MATBG.

Questions:

1. A key issue as all the referees mentioned is the reproducibility of the physics. Despite the new sample B in this manuscript, the asymmetric Fraunhofer was not presented in this new device. There is only a SQUID-like interference pattern which the authors attributed to the low carrier density in the CI state. Therefore, this new sample doesn't provide any supporting results for the symmetry-breaking mechanism as proposed by the authors. This is the key issue for this manuscript.
2. As the referee pointed out that a single device result will need strong support from a concrete theoretical model. Here the authors explained the asymmetric Fraunhofer and magnetic field hysteresis using different assumptions. In the prior case, an unusual current density distribution has to be made together with valley polarization and orbital magnetization. Another assumption is the half-filling in the CI state. Overall, it seems to be very hard to produce the observed Fraunhofer for which one has to realize several requirements for such a scenario. For the hysteresis of the interference pattern, spin, and valley polarization are needed. The linking between the two cases is relatively weak.

3. Is possible the peculiar current density distribution of device A from an edge or disorder effect?
4. Possible effect of the in-plane field? While the perpendicular field is applied to the sample with the amount of a tenth Tesla, is it possible a small misalign of the field resulting in an in-plane field causing the symmetry breaking or opening up the valley degeneracy?
5. Device B did not show a zero-resistance state as shown in Fig. S6 B. This could be a simple reason for no superconducting current through the sample. The interference pattern from Fig. 2 d is the result of two weak links due to edge channels. On the other hand, does device B's period of interference pattern consist of the flux quanta?

Minor:

1. Figure 1d X-axis unit bracket became the upper case.
2. Figure g Y-axis is shifted up? Otherwise, it seems to become negative resistance

Reviewer #3 (Remarks to the Author):

Questions:

Thanks to the authors' reply. Overall, I am fine with some questions from the earlier manuscript. However, before publishing, there are still some issues.

Comment:

1. One of the questions from the referee is whether the author should observe the quantum anomalous Hall effect due to the valley polarization with a Chern number of -2 for the correlated insulating state. This effect could be greatly lifted due to the geometry effect. In a narrow junction geometry, the R_{xx} and R_{xy} often have severe mixing. Indeed, if one considers the Hall geometry, the anomalous Hall effect may be expected. However, it will be hard to see in a gated junction geometry. See the following reference: Williams, J. R., Abanin, D. A., DiCarlo, L., Levitov, L. S., & Marcus, C. M. (2009). Quantum Hall conductance of two-terminal graphene devices. *Physical Review B*, 80(4), 045408.

We thank the reviewer for the comment and the interesting reference.

2. The new device B has relatively weak superconductivity as shown in the SI. The resistance doesn't reach zero at zero fields. This may be hard to reproduce the CI state in device A. It is also the reason that interference shows a SQUID pattern due to the poor superconducting properties of MATBG.

We thank the reviewer for the comment. The fact that the superconductivity does not show a zero-resistance state suggests that the superconducting region does not percolate completely across the two measuring voltage probes, however we are convinced that there is fully developed SC in the junction region. This has been understood before due to a local twist angle inhomogeneity (Uri, A., (2020). *Nature*, 581(7806), 47–52). However, we do not agree the SQUID-like pattern is linked to this. However, we are convinced of the existence of an uninterrupted SC state in the junction region. This can be understood from Fig. S6, which clearly shows that when measuring in the SC/SC/SC case (Fig. S6e), no Josephson oscillations are observed, proving the homogeneity of the superconducting state. Only when tuning the weak link region closer to $\nu = -2$ do we observe the SQUID-like pattern (Fig. S6f).

Questions:

1. A key issue as all the referees mentioned is the reproducibility of the physics. Despite the new sample B in this manuscript, the asymmetric Fraunhofer was not presented in this new device. There is only a SQUID-like interference pattern which the authors attributed to the low carrier density in the CI state. Therefore, this new sample doesn't provide any supporting results for the symmetry-breaking mechanism as proposed by the authors. This is the key issue for this manuscript.

We thank the reviewer for the important comment and we will try to clarify the importance of the second device here. In the revised version of the manuscript, following the comments of two of the referees in the original manuscript (including referee 3), we note how in device A edge states play an important role in the interpretation of the data. In the case of having a valley polarized state as being responsible for the unconventional behavior of device A, one would expect edge states to carry a fair amount of the supercurrent. This agrees with the experimental

data, where device A shows a revival of the oscillations (Fig. 2a), denoting the presence of edge states. In device B, at the same filling close to $\nu = -2$, the Fraunhofer pattern shows even clearer SQUID-like oscillations. This is the reason why in the second version we have modified the manuscript to remark the presence of edge states in both devices, while the asymmetry and the magnetism are only present in device A. To highlight this, we have added a sentence in **line 148**: “For both devices we observe how in the SC/CI/SC configuration edge states play an important role carrying the supercurrent”. Furthermore, in the effective tight binding theoretical model (sections J to N of the SI), we also see this behavior. As is shown in section N of the SI, in order to observe the asymmetry in the model, we need to have a combination of bulk and edge conduction. When putting the bulk fully insulating and having only edge conduction, we recover an almost perfect SQUID pattern as shown in Fig. S18 and explained in the SI lines 583-590. This is in great agreement with the experimental observation, as shown with the current density distribution analysis in Fig. 2.

Therefore, although we agree with the referee that sample B does not fully reproduce the whole dataset of device A, we believe it is a good support to show that the presence of edge states plays an important role in the Josephson junction around the CI at $\nu = -2$, which supports the hypothesis of this state being valley polarized with Chern number $C = -2$. We believe that having both samples highly strengthens the paper and can help to get a deeper theoretical understanding of the MATBG system and, in particular, to the state at the CI at $\nu = -2$, which ground state is still under debate, both theoretically and experimentally (Park, J. M. *et al. Nature*, 592(7852), 43–48; Yankowitz, M. *et al. Science*, 363(6431), 1059–1064; Potasz, P., Xie, M., & MacDonald, A. H. (2021). *Physical Review Letters*, 127(14); Tseng, C.-C. *et al.* (2022). *Nature Physics*, 18, 1038–1042).

2. As the referee pointed out that a single device result will need strong support from a concrete theoretical model. Here the authors explained the asymmetric Fraunhofer and magnetic field hysteresis using different assumptions. In the prior case, an unusual current density distribution has to be made together with valley polarization and orbital magnetization. Another assumption is the half-filling in the CI state. Overall, it seems to be very hard to produce the observed Fraunhofer for which one has to realize several requirements for such a scenario. For the hysteresis of the interference pattern, spin, and valley polarization are needed. The linking between the two cases is relatively weak.

We thank the reviewer for the question and we will try to clarify here. As we explain in the main text we build an effective tight binding model with a valley polarized state with Chern number $C = -2$ as the weak link of the junction. We observe that in this case we can simulate the asymmetry observed in the experiment and that this goes away when we remove the terms which break the C_2T symmetry (Fig. S17). Furthermore, we see how just by making the bulk more insulating we can get a SQUID-like pattern, as we observe experimentally for device B (Fig. S18). Therefore, the main features of both devices can be captured within the same model. We have emphasized this in the text by modifying the sentence in **lines 195-198** to: “We can then simulate the behavior of device B by setting the chemical potential into the gap of the CI, equivalent to having a more insulating bulk (see SI), in which case the current is solely carried by edge states, consistent with the results shown in Fig. 2. Therefore, the main features of both devices can be captured within the same model.”

Furthermore, we have made a phenomenological analysis considering the current density distribution of the data in order to understand the role of edge states. From this phenomenological analysis we find that in order to observe the asymmetric pattern of device A we need to have a combination of bulk and edge conduction, combined with a broken time reversal symmetry in the edges. However, we find that the pattern of device B is characterized by having only edge state conduction with an insulating bulk. These features are in agreement with the effective tight binding model explained above.

Finally, we note that the low temperature hysteresis cannot be explained by the effective tight binding modelling. In the original text we explained this effect by considering that we had a further spin polarization at low fields. We want to emphasize how in the new version we added a new paragraph in the discussion (**lines 230-244**) to explain that we cannot conclude the origin of this signal. We have kept the spin model in the SI (Fig. S19), because we still consider a partial spin polarization as one of the scenarios. However, we also consider other scenarios, as we explain in the discussion. To clarify this, we have emphasized it in the **SI section N in lines 591-593**: “Finally, we discuss the feature of the Fraunhofer pattern in the presence of spin magnetism. As we discussed, the spin magnetism is one of the possibilities to give rise to the data in the low-temperature range ($T < 800$ mK) according to the measured Fraunhofer patterns.”.

We agree with the referee that explaining all these signals is a very complex task and we actually cannot explain them completely with the present data and theoretical knowledge, that is why we added a sentence in **line 228** stating that: “Therefore, a further theoretical explanation is needed to explain these lower T features which have a clear distinct behavior.”, and in **line 242** to conclude as: “... a definite proof of the origin of these signals cannot be drawn from the present study”.

3. Is possible the peculiar current density distribution of device A from an edge or disorder effect?

We thank the reviewer for the interesting comment. However, we do not believe the unconventional Fraunhofer pattern of device A to come from just a disorder effect, since it appears only for a certain range of carrier densities when tuned very close to the CI state at $\nu = -2$. As we show in Fig. 1e or in Fig. S7, we can gate in and out of this Fraunhofer pattern. In the case of the effect arising from some edge effect defect or some disorder, it should be independent of the carrier density of the weak link. For example, in that case one would expect Fig. 1e left to also show asymmetries, while it does not.

4. Possible effect of the in-plane field? While the perpendicular field is applied to the sample with the amount of a tenth Tesla, is it possible a small misalign of the field resulting in an in-plane field causing the symmetry breaking or opening up the valley degeneracy?

We thank the reviewer for the comment. Although the effect of in plane field could be interesting, we do not believe any effect from misalignment could influence the data. In our set-up the sample is loaded in a stage which stays perpendicular to the magnetic field, although due to the sample mounting a misalignment is always expected. However, this misalignment is rather small, being in the order of $1^\circ - 2^\circ$, as is often seen in vector magnet experiments. We have actually measured such misalignment in this device in a vector magnet setup (the procedure is explained below), and similar numbers are found in literature (see for example the

supplementary materials of Cao, Y., *et al. Science*, 372(6539), 264–271). If we assume that the original setup has a misalignment of up to 2° , that would mean we would induce an in-plane field B_{\parallel} of $\sim 2.2\%$ of the applied out of plane field B_{\perp} . In this case, when having a $B_{\perp} = 10$ mT, we would get a consequent parallel field $B_{\parallel} \approx 220 \mu\text{T}$.

In MATBG it has been seen that due to the large unit cell of MATBG of $\lambda \sim 14$ mT, a substantial magnetic flux can be trapped between the two layers in the area formed between the unit cell length and the separation of the layers $\delta \sim 0.3$ nm, $A = \lambda \cdot \delta$. This flux can effectively lift the valley degeneracy due to a momentum shift between the two layers and is attributed for the suppression of the superconductivity in MATBG in the presence of an in-plane field (Qin, W., & Macdonald, A. H. (2021). *Physical Review Letters*, 127(9); Cao, Y. *et al.* (2021). *Science*, 372(6539), 264–271). The characteristic energy shift is given by $E = g\mu_B B_{\parallel}$, where $g = 2$ and μ_B is the Bohr magneton and effects are seen when applying in-plane fields in the order of 1 T. For the range of fields discussed above (hundreds of μT), we could expect energy shifts of < 100 neV. Considering that the observed effect survives up to 750 mK in device A (see Fig. 4), one could neglect the effect of such in plane fields to have any effect. The energy scale to have a broken symmetry that survives to 750 mK would be in the order of $E = k_B T_c \approx 65 \mu\text{eV}$, more than two orders of magnitude larger than the expected from a parallel field effect due to misalignment of the sample.

Fig. R1. Aligning vector magnet procedure. In order to know the angle misalignment in a vector magnet setup we find a feature which is highly sensitive to an out-of-plane field and calibrate it when applying different in-plane fields. In the case of MATBG we go to the edge of the superconducting dome which is very sensitive to the out-of-plane field (B_x in this case) as shown in Fig. a. Then we apply different in plane fields and follow the evolution of the sample behavior (see Fig. a). Then we apply different values of the in-plane fields (B_y and B_z in this case) and extract the minima as explained in a. Finally, we fit these minima with a plane equation $B_x = aB_y + bB_z + c$, to obtain the real space plane of the sample, as shown in Fig. b. From this plane we can extract the angle of the sample. In this case we find a tilt angle of $\approx 1.5^\circ$ with respect to the B_y - B_z plane.

5. Device B did not show a zero-resistance state as shown in Fig. S6 B. This could be a simple reason for no superconducting current through the sample. The interference pattern from Fig. 2 d is the result of two weak links due to edge channels. On the other hand, does device B's period of interference pattern consist of the flux quanta?

We thank the reviewer for the question. The fact that the superconductivity does not show a zero-resistance state suggests that the superconducting region does not percolate across the two measuring voltage probes. This has been understood before due to a local twist angle inhomogeneity (Uri, A., (2020). *Nature*, 581(7806), 47–52). As the referee points out, we argue in the text that the pattern is due to the supercurrent being carried by edge states. The period of the oscillation has a good agreement with the expected flux quanta. The period of oscillation is of ≈ 5 mT, as can be extracted from Fig. 2d. The width of our Hall bar is $w \approx 800$ nm. Using the 2D Josephson junction formalism explained in section G of the SI, we can expect a period of $\Delta B_{2D} \approx 1.8 \Phi_0/w^2 = 5.6$ mT, in very good agreement with the measured data.

Minor:

1. Figure 1d X-axis unit bracket became the upper case.
2. Figure g Y-axis is shifted up? Otherwise, it seems to become negative resistance.

We thank the reviewer for pointing out these two mistakes. Both of them have been now corrected.

REVIEWERS' COMMENTS

Reviewer #3 (Remarks to the Author):

Thanks to the authors' detailed explanation. My concern and questions about the manuscript have been addressed. I believe that the manuscript is ready to be published in Nature Communication.